# Long-Term Stability of a RAFT-Modified Bulk-Fill Resin-Composite under Clinically Relevant versus ISO-Curing Conditions

**DOI:** 10.3390/ma13235350

**Published:** 2020-11-25

**Authors:** Niklas Graf, Nicoleta Ilie

**Affiliations:** Department of Conservative Dentistry and Periodontology, University Hospital, Ludwig-Maximilians-University, 80336 Munich, Germany; niklas.graf@web.de

**Keywords:** bulk-fill resin composite, RAFT-polymerization, long term-stability, curing conditions, accelerated aging, fractography, Weibull analysis, three-point bending test, depth-sensing indentation

## Abstract

The addition of RAFT (reversible addition-fragmentation chain transfer) agents to the matrix formulation of a bulk-fill resin composite can significantly decrease the required curing time down to a minimum of 3 s. Evaluating the long term-stability of this resin composite in relation to varied curing conditions in an in-vitro environment was this study’s goal. Specimens were produced according to either an ISO or one of two clinical curing protocols and underwent a maximum of three successive aging procedures. After each one of the aging procedures, 30 specimens for each curing condition were extracted for a three-point bending test. Fragments were then stereo-microscopically characterized according to their fracture mechanism. Weibull analysis was used to quantify the reliability of each aging and curing combination. Selected fragments (*n* = 12) underwent further testing via depth-sensing indentation. Mechanical values for either standardized or clinical curing were mostly comparable. However, changes in fracture mechanism and Weibull modulus were observed after each aging procedure. The final procedure exposed significant differences in the mechanical values due to curing conditions. Curing conditions with increased radiant exposure seemingly result in a higher crosslink in the polymer-matrix, thus increasing resistance to aging. Yet, the clinical curing conditions still resulted in acceptable mechanical values, proving the effectiveness of RAFT-polymerization.

## 1. Introduction

The wide variety of bulk-fill resin-based composites (BF-RBC) currently in the dental market offer clinicians an efficient and accelerated way, compared to conventional resin-based composites, of directly restoring cavernous teeth. A higher depth of cure, generally achieved by a decreasedfiller volume percentage, a smaller filler-matrix interface and fewer color pigments, raises the maximum increment height above the conventional 2 mm threshold [1]. Thus, clinicians are theoretically able to use fewer increments and less frequent light exposures to finish a restoration. Despite the latter, each curing duration is still advised to be at least 10–20 s to adequately polymerize most of these BF-RBC’s [1].

Different approaches have been made to further reduce the required curing times. The usage of high-powered light-curing units (LCU), capable of emitting adequate irradiance and radiant exposures in a shorter amount of time through the ongoing improvement of light-emitting diode devices, was one of them [2]. Another was to optimize the photo-induced radical initiation reaction of the photoinitiators. Replacing camphorquinone and the associated tertiary amine, a Norrish type-II initiator, with a Norrish type-I initiator, i.e., monoacylphosphine oxides (Lucirin-TPO^®^) [3] or benzoyl-germanium derivates (Ivocerin^®^) [4], led to an increased generation of free radicals. An increase in initiating radicals in turn leads to a higher photoactivity and more starting points of propagating radical polymerization, theoretically leading to an optimized polymer network. However, pairing these new photoinitiators with the more translucent BF-RBC’s still did not make curing times lower than 10 s a reality. Many studies concluded that longer exposure times still produced higher depths of cure and/or higher mechanical properties in commercial bulk-fills [5,6,7]. In this regard, another study concluded that an experimental Lucirin-TPO^®^-based composite in combination with a specifically tailored LCU employing curing times of ≥3 s led to higher degrees of conversion and mechanical properties than for a conventionally cured camphorquinone-based experimental composite [8]. Even though this study only used experimental composites, it still highlighted the effectiveness of a closed-system approach in-regards to LCU and photoinitiator.

Recently, a new method of polymerization has found its way into dentistry. The so-called reversible addition-fragmentation chain transfer polymerization (RAFT-polymerization) has seen numerous applications in polymer sciences since its inception in 1998 [9]. By using mostly dithioester compounds, so called RAFT-agents, new radicals are generated as a by-product each time a monomer undergoes addition into a polymer network [9]. This mechanism has seen few applications in dentistry so far. The subject material of this study, Tetric PowerFill (TePF), is one of the exceptions to this. A recent study has shown promising results of the material in regards to degree of conversion when employed both in ultra-fast and conventional photopolymerization in comparison to an already commercially established BF-RBC [10]. Limited data on this new type of composite is available as of now, especially when considering the performance in an in-vivo environment.

Accelerated lab-side aging can be an important tool in predicting long-term stability of a restorative material in a clinical environment. Simulated changes in temperature and storage medium can critically impact the integrity of the BF-RBC and consecutively change its mechanical properties and morphological features [11]. Changes in polymerization time can also critically alter a composite’s mechanical stability and inadequately polymerized specimens are generally more susceptible to the effects of the oral environment [12]. These changes in measured values and morphology can be related with the help of fractography. Examination of the fracture surface variation can be related to the mechanical properties of a cohort of test specimens, leaving room for interpretation about the effects of accelerated aging [13].

As outlined above, the information about the long-term behavior is vitally important for a material like TePF, which not only employs a polymerization mechanism, rarely used in dentistry, but also promises to lower irradiation times significantly. Still, the legitimate relevance of optimal versus clinical curing conditions needs to be addressed. Therefore, the null hypotheses were:
(1)Accelerated aging does not alter the mechanical properties or reliability of the material,(2)Optimal curing, as stipulated in standards [14], does not significantly change the measured properties or reliability in comparison to clinically relevant curing times and exposure conditions,(3)Accelerated aging does not change the fracture mechanism of the test specimens,(4)Variation in curing time and curing direction do not affect the fracture surface or mechanism of the test specimens.

## 2. Materials and Methods

A commercially available sculpt-able BF-RBC (Tetric PowerFill, TePF, Shade IV A, LOT# XZ1097, Ivoclar Vivadent, Schaan, Liechtenstein), capable of RAFT-polymerization, was studied by alteration of curing time and curing direction (ISO or 20 s top and bottom, 10 s top and 3 s top of continuous curing) and aging conditions (24 h storage in artificial saliva, followed by thermocycling and subsequent storage in a solution of 75 vol.% ethanol/25 vol.% distilled water for 7 days). Consequently, a total of nine groups (*n* = 30) for each aging and curing method were examined. The progressive storing conditions, 24 h storage in artificial saliva, 24 h storage in artificial saliva followed by thermocycling, and 24 h storage in artificial saliva followed by thermocycling and successive storage in alcohol for 7 days, will be referred to as step one, step two and step three respectively (see Figure 1).

The test slabs were manufactured by condensing the BF-RBC into a Teflon mold (internal dimension of 16 mm × 2 mm × 2 mm in between two glass panels separated by a Mylar-strip (Matrix-strips, REF#143274, Orbis Dental Handelsgesellschaft GmbH, Münster, Germany)). Curing was then performed according to either ISO-standards [14] or 10 s and 3 s overlapped top polymerization only, to simulate clinical situations. Operating the manufacturer-specified light-curing unit (LCU, Bluephase PowerCure, Ivoclar Vivadent, Schaan, Liechtenstein), three different curing modes were used for the respective testing groups (20 s “high”, 10 s “high” and 3 s “flash”). Furthermore, the specimen’s surface, which was polymerized at first, will be referred to as “top”. The opposite surface will be referred to as “bottom”. The test objects were then ground down with silicon carbide sandpaper (P1200, Hermes, EXAKT Advanced Technologies GmBH, Norderstedt, Germany) to even out any protrusions or disturbing edges.

The samples were then stored in artificial saliva for 24 h at 37 °C inside an incubator of which one third was removed for reference testing. The rest underwent further aging (thermocycling for 10,000 cycles at 30 s in 5 °C to 55 °C distilled water per temperature; Thermo Scientific, Waltham, MA, USA) of which one third was again extracted for evaluation. The remaining samples were placed in a solution of 75 vol.% Ethanol/25 vol.% distilled water and stored for 7 days at 37 °C before final analysis.

### 2.1. Three-Point Bending Test

Every sample (*n* = 30) was loaded until failure in a three-point bending test, with 12 mm in between the supports conferring to NIST No. 4877, using a universal testing machine (Z2.5, Zwick/Roell, Ulm, Germany). Flexural strength and flexural modulus were recorded with the crosshead directed towards the top of the blocs at a speed of 0.5 mm/min. The flexural strength was calculated, using the formula: σ = 3Fl/2bh^2^, where *F* is the maximum load applied, *l* is the distance in between supports, *b* is the width, and *h* is the height of each specimen. According to Hooke’s law, the flexural modulus was calculated as the slope during the linear portion of the loading process. During testing, all specimens were immersed in distilled water at room temperature. The fragments were then carefully extracted and prepared for further examination.

### 2.2. Fractographic Analysis

The surface quality at the point of fracture was examined using an optical microscope (Stemi 508, Carl Zeiss GmbH, Göttingen, Germany). At that point, the fracture origin was determined and categorized into one of four distinctive groups (sub-surface, edge, corner or plain fracture). Each surface was then photographed and documented using a microscope camera extension (Axiocam color 305, Carl Zeiss GmbH, Göttingen, Germany). Three specimens for each fracture mode were then selected and sputtered with a gold-palladium coating for scanning electron microscopy (Zeiss Supra 55VP, Carl Zeiss GmbH, Göttingen, Germany). Thus, surface morphology, fracture origin and crack propagation path underwent further examination.

### 2.3. Depth-Sensing Indentation Test

Twelve fragments of every testing-group were then used to determine the micro-mechanical properties of the RBC. Prior to testing, fragments were fixed onto an acrylic-slide, so that the top (*n* = 6) and bottom (*n* = 6) of the fragments could be ground down by 200 µm and polished in an automatic grinder (EXAKT 400CS Micro Grinding System, EXAKT Advanced Technologies GmBH, Norderstedt, Germany) with the help of silicon carbide sandpapers (Hermes, EXAKT Advanced Technologies GmBH, Norderstedt, Germany) according to a defined protocol (first: 150 μm of reduction at P1200, second: 50 μm of reduction at P2500, third: polishing at P4000). The measurements were performed utilizing an automatic micro hardness indenter (Fischerscope H100C, Fischer, Sindelfingen, Germany) with six randomly distributed indentations on each fragment and specimen side, resulting in 72 values for each testing-group. Conferring to ISO standards [15], controlled force was achieved by a constant load increase and load decrease starting at 0.4 mN up to 500 mN. The universal hardness, defined as the test-force divided by the apparent area of indentation, was recorded and the indentation modulus calculated from the slope of indentation at maximum force. Due to the implementation of a conversion factor, based on multiple measurements provided by the manufacturer, Vickers’ hardness for each measurement could in turn be calculated and displayed in the software. Creep was measured as the change in deformation under the persistent load.

### 2.4. LCU Characteristics

The irradiance, radiant exposure and spectral distribution of each light curing mode of the LCU used were examined using a NIST-referenced USB4000 Spectrometer (MARC-RC *(Managing Accurate Resin Curing)* System, Bluelight Analytics Inc., Halifax, NS, Canada). Six measurements were performed for each specific curing mode to rule out any discrepancies regarding the light curing portion of the manufacturing process. During measurement, the light guide, 9 mm in diameter, was placed directly on top of the sensor, 3.9 mm in diameter, to prevent any additional scattering of emitted light. The results were again compared to the manufacturer’s data to double-check the integrity of the LCU.

### 2.5. Statistical Analysis

Based on the conceptualization of the study, the dependent variables were flexural strength, flexural modulus, Martens’ hardness, Vickers’ hardness, indentation modulus, Creep, and fracture mode. The independent variables were thus aging and curing mode. A Kolmogorov–Smirnov test was utilized to establish normal distribution of the test results. Equality of variance was determined by way of Levene’s test. Multiple student’s *T*-tests determined significant differences in values measured through the depth-sensing indentation test, in relation to the measurement surface (top-bottom). A multivariate analysis explored and quantified the effects of aging and curing on the measured parameters. Post hoc-test comparisons of the results were established by a one-way analysis of variance (ANOVA) and Tukey honestly significance difference (HSD) using IBM SPSS (Version 26.0, Chicago, IL, USA). Additionally, the reliability of each material was examined using a Weibull analysis. The Weibull model describes the probability of failure for brittle materials at uniform stress using the formula: *P_f_(σ_c_)* = 1 − exp [−(*σ_c/_σ*_0_)^m^].

Where *σ_c_* is the measured strength, *σ*_0_ is the characteristic strength at probability of failure *P_f_ (σ_f_)* = 0.63, and *m* is the Weibull modulus.

The double logarithm of this expression: lnln [1/1 − *P_f_*(*σ_c_)*] = *m*ln(*σ_c_)* − *m*ln*(σ*_0_*)* results in a straight line. The upward gradient of that straight line in turn represents *m. R*^2^ exhibits the fit of variance of the observed data towards the projected ideal linear function.

## 3. Results

The variables measured in the three-point bending tests and the corresponding Weibull moduli are illustrated in Table 1 and Table 2.

In general, progressive aging led to a significant decrease in all properties, measured by the three-point bending test, for each curing group with a more pronounced effect on flexural modulus. In terms of curing, a statistically significant difference in between groups was only observed after the final step of aging with respect to flexural strength. In contrast, the flexural modulus of the different curing approaches differed significantly after the first step of aging. On the other hand, the groups, in relation to mode of curing, did not differ after the second step of aging. The final step of aging followed the same trend as with the flexural strength parameter. The effective influence of the variation of curing and aging, i.e., partial η_P_^2^-values, were verified by way of multivariate analysis of variance. Aging had the greatest effect on the measured properties. It had a corresponding effect on flexural modulus (η_P_^2^ = 0.828) and strength (η_P_^2^ = 0.579). However, curing had a greater effect on the flexural modulus (η_P_^2^ = 0.175) compared to flexural strength (η_P_^2^ = 0.055). Several trends can be observed from the Weibull analysis and graph. Firstly, in comparison, the Weibull moduli of the ISO- and 3 s-curing group were very similar for each aging step. Secondly, all groups showed a very similar Weibull modulus after aging step three (see Table 1). Lastly, a general shift of the distribution to the left commenced with increased aging (see Figure 2).

Identification of the fracture modes resulted in an overwhelming majority of sub-surface fractures (78.5%), with corner fractures in second (8.1%); the less-brittle fracture mode, titled plain, in third (7.8%); and with edge fractures being the least likely fracture mode (5.6%). However, the amount of plain fractures did increase with progressive aging (see Figure 3).

A Chi-squaredtest showed no significant dependency of curing mode and fracture mode (*p* = 0.967), yet it did expose a significant relation of aging and fracture mode (*p* < 0.05). A One way-ANOVA with subsequent Tukey’s post-hoc test showed a significant decrease in flexural strength and flexural modulus when fractures occurred in the plain fracture mode in comparison to the other identified mechanisms (*p* < 0.05). The other fracture modes contrarily formed a homogeneous sub-set for both flexural strength (*p* = 0.472) and flexural modulus (*p* = 0.083). A representation of each fracture mechanism is shown in Figure 4A–D.

The results of the depth-sensing indentation test are visualized in Table 3, Table 4, Table 5 and Table 6.

Results of the student’s *T*-test’s showed significant differences for the majority of measurements taken at either the top or bottom surface (*p* < 0.05), with a more pronounced effect after aging. As a consequence, top and bottom surface values will be addressed separately. Increased curing time mostly led to increased micro-mechanical properties, with the ISO-curing group achieving the highest values for indentation modulus, Martens’ and Vickers’ hardness at both the top and the bottom of the test subjects. Yet, after thermocycling, the 3 s-curing group scored the highest values for the aforementioned parameters at the top-surface. The different curing conditions were rarely completely homogeneous in relation to a respective aging mode. A tendency for the 3 s-group to either confer with the 10 s-curing group or stand completely isolated in terms of homogeneous groups was detected, as evidenced by the Tukey’s post-hoc test. Creep on the other hand remained mostly unaffected by the variation in curing time/direction. Artificial aging led to a specific pattern across all curing groups regarding indentation modulus and the hardness values. Step two led to a significant decrease in indentation modulus while the hardness values either decreased or remained unchanged, at both surfaces. After step three of aging, all mentioned variables experienced an increase, when compared to the prior aging-step, at the topsurface. An increase in the bottom values was also observed for the ISO- and 10 s-curing groups, yet the bottom of the 3 s-curing group did show a sharp decrease. Standard deviations for the variables mentioned above also increased noticeably with successive aging. Meanwhile, Creep increased after the second step of aging and remained constant for both surfaces after the final step of aging. The exception to this was the bottom surface of the 3 s-curing group after aging step three. Again, η_P_^2^-values were used to quantify the effective influence on each of the variables. Curing altered Martens’ hardness (η_P_^2^ = 0.045) and Vickers’ hardness (η_P_^2^ = 0.063) as well as indentation modulus (η_P_^2^ = 0.05) in a similar fashion. Creep was affected the most (η_P_^2^ = 0.287). Aging again influenced Creep the most (η_P_^2^ = 0.415), whilst the indentation modulus was also noticeably modified (η_P_^2^ = 0.257). Aging produced a comparable response in Martens’ (η_P_^2^ = 0.058) and Vickers’ hardness (η_p_^2^ = 0.037). Position had a coinciding effect on the hardness values (HM η_P_^2^ = 0.111; HV η_P_^2^ = 0.103) and indentation modulus (η_P_^2^ = 0.128). Lesser consequences were calculated for Creep (η_P_^2^ = 0.035).

Characterization of the used LCU reported a mean irradiance of 3451.96 mW/cm^2^ (8.13) for the curing mode *flash* (3 s) and 1383.78mW/cm^2^ (7.04) for the curing mode *high* (10 s and 20 s). The radiant exposure received by each specimen was measured at 10.24 J (0.05) and 13.75 J (0.04) for the 3 s- and 10 s-curing mode (top surface only) while the ISO-curing mode (both top and bottom surface) amounted to 55.29 J (0.16). The emission spectrum of the LCU showed two peaks, one at 410 nm (violet wavelength range) and the other at 450 nm (blue wavelength range).

## 4. Discussion

The goal of in-vitro aging procedures is to mimic clinical conditions as best as possible. The information gathered can predict the longevity of the material in an oral environment, while variation in curing time, i.e., ISO versus clinical curing times, can also be investigated. This is particularly important for TePF since it can be used in an open-system approach or in connection with the manufacturer-specified LCU. According to the manufacturer’s data, the material in question may be adequately photo-polymerized in combination with any LCU, emitting an irradiance up to 2000 mW/cm^2^. Yet, note that the 3 s-polymerization mode is exclusively limited to the high-intensity LCU used in this study [16]. One needs to be aware that the radiant exposure (irradiance x curing-time; RE) received by a restoration, using the 3 s-curing mode, is the lowest compared to the 10 s- and 20 s-curing modes. The RE of the curing mode of the 3 s-curing mode amounts to 75% compared to RE of the 10 s-mode. The difference in RE is even more defined for the ISO-curing mode since the ISO-method provides twice the exposure of the 20 s-curing mode. Consequently, the RE received in the 3 s-mode resembles about 18% of the RE-received under ISO-conditions. After all, this means that a fewer amount of total photons reach the restoration, particularly the bottom portion. Thus, less radicals are being cleaved off the photoinitiators, which leads to fewer hot spots of propagating polymerization [17]. This makes the choice of photoinitiator all the more important. To guarantee a sufficient starting reaction, TePF employs two different photoinitiators: camphorquinone, coupled with a tertiary amine, as well as a benzoyl germanium derivate, Ivocerin^®^ [16]. Photo-induced cleavage of camphorquinone results in one active radical being generated, yet benzoyl germanium derivates produce a minimum of two radicals, thus increasing photoactivity [4]. The emission-spectrum of the manufacturer-specified LCU echoes the absorption peaks of those two compounds closely, in order to induce enough starting radicals even when employing ultra-fast curing modes.

A recent study evidenced that, when employed in combination with the manufacturer-specified LCU, 3 s-curing yields comparable results to the 10 s-curing mode in terms of depth of cure, degree of conversion and flexural strength [10]. This is in line with the findings of this study, even though artificial saliva, in contrast to distilled water, was used as the primary storage medium. A long-term study on sorption and solubility of bulk-fill and conventional resin composites by Alshali et al. demonstrated no significant differences for Tetric Evo Ceram Bulk-Fill, the BF-RBC on which TePF’s matrix-composition is based on [16], in terms of these two storage media [18]. The aging steps had a highly significant effect (η_P_^2^ = 0.579) on the flexural strength of the material for all curing groups. The third step of aging, however, affected the 10 s- and 3 s-group more drastically than the ISO-group, indicating a higher resistance to the aging conditions (see Table 1). Since one and the same material was used in all groups, the measured differences can be assigned to the organic matrix. As the matrix-filler interface remained consistent across the whole study, the differences in flexural strength give important clues about the changes in the polymer network. Additionally, a meta-analysis by Heintze et al. stated a significant correlation between a decrease in both flexural strength and clinical index for several studies with comparable specimen dimensions and aging protocols [19]. This accentuates how the progression from aging step two to three might not only be highly detrimental to the polymer network in-vitro, but also the most relevant in clinical terms.

The trend mentioned above is confirmed by the flexural modulus, which turned out to be more sensitive to both, as evidenced by the higher η_p_^2^-values. Even if the differences can only be seen in decimal places, they are significant 24 h post-polymerization, indicating a possible difference in polymer crosslinking (see Table 2). Interestingly, all groups formed a homogeneous sub-set after step two of aging. The hint at possible changes in polymer structure was solidified with subsequent aging, as the 10 s- and 3 s-group again evidenced the highest decrease in their flexural modulus, while the ISO-group again seemed more resistant to this type of aging.

The variation of RE, while not affecting the overall degree of conversion [10], did cause changes in the polymer crosslink density. Alcohol is an extreme solvent of dental composites as it can easily penetrate the less densely crosslinked areas inside the polymer network, leach out any unreacted monomers, and thus weaken the structural integrity of the organic matrix [11,20,21]. As already mentioned, the 10 s- and 3 s-group receive less photons due to the lower amount of RE, and fewer radicals are generated by photoactivation alone. Henceforth, the polymer-chains grow longer since it is less probable for two propagating polymer networks to cross-react with another [22,23]. The presence of filler particles might also inhibit a propagating polymer chain [6], forcing it to either form a loop with the other end of its own polymer chain or stopping the addition of further monomers all together, resulting in a pendant reactive group. The degree of conversion of the material might not bear any sign of this, because only the relative conversion of C=C double-bonds into C–C single-bonds is measured, not the monomer conversion or the chemical crosslink density [24]. The aforementioned issue of possible lower crosslink was addressed by modifying the matrix to enable RAFT-polymerization. The inclusion of β-allyl sulfones, an additional fragmentation chain transfer (AFCT) reagent, to the composition of the material [16], tries to combat this problem to an extent by enabling the photo-radical to start a chain-transfer reaction. The photo-radical forms a single-bond with the allyl part of the sidechain, which ultimately leads to fragmentation of the sulfone compound, carrying a free electron, from the methacrylate monomer. This fragmented sulfone compound might in turn engage in monomer addition or another chain-transfer reaction, which could in turn lead to polymer crosslinking [25]. Nonetheless, the problem of fewer initiating photo-radicals still remains, as evidenced by the fact that the 10 s- and 3 s- curing protocols produced the lowest flexural strength and modulus values after the final step of aging. This was due to the fact that in the three-point bending test, the bottom side of the specimen was deliberately placed in the tensile zone and thus had a decisive effect on the flexural strength. The ISO-curing group was generally less affected, probably due to the optimal configuration of the polymer network as a result of the maximum amount of RE received [55.28J (0.16)] and polymerization from both sides. However, an ISO-cured specimen is polymerized to the amount, which is not attainable clinically. Realistically, a clinical restoration will never be cured directly from the bottom. Both shorter curing times still resulted in acceptable values compared to contemporary bulk-fill composites, serving as a proof of concept of the RAFT-modification of the monomer formulation. Especially when one considers the small differences in absolute value (see Table 2) [26].

More insight into the failure of the specimens is provided by way of Weibull and fractographic analysis. Critical flaws inside a specimen can be interpreted visually with the help of fractography, while the Weibull analysis can quantify the flaw population. Both methods can be used in this study, since a large number of test pieces (*n* = 30) is available. Larger quantities of test subjects help increase the reliability of the Weibull analysis [27]. A high Weibull modulus, i.e., the slope of the graphs presented in Figure 1, represents a low variation of flaw size and/or type. A high fit of the data distribution towards a linear function (partly the R^2^-determinacy of the Weibull graph) is indicative of a dominant failure mode [27]. In this study, the fracture modes were categorized based on the morphology of the entire fracture surface. A sub-surface fracture represented a void or inhomogeneity below the specimen’s surface. Corner and edge fractures, however, were classified as a surface void or pores, located at the very surface of the specimen. These three fracture types generally showed very distinct brittle features, like large compression curls, smooth fracture mirror and distinct Hackle lines (see Figure 4A–C). The plain fracture mode in contrast showed fewer of these features. Very planar fracture surfaces, barely identifiable fracture mirrors and less-pronounced Hackle lines were observed stereo-microscopically. This is indicative of a low-stress fracture, which is in line with the results of the ANOVA, linking plain fractures to a lower flexural strength and modulus [13]. The first aging step produced the highest population of the more brittle fracture modes (see Figure 3), with the fewest amount of plain being recorded overall. As expected, this leads to a general shift of the data distribution to the right of the Weibull graph. Weibull moduli for all curing groups were high (see Table 1), yet the 10 s-curing group scored the lowest out of all groups. When looking diligently at the graphical data of the group, one notices an outlier located very far on the left of the Weibull spectrum. The photographed fracture surface shows a massive corner defect, which was probably covered by a thin layer of composite at the beginning of the test. This layer could have either fractured off while loading or at the very start of the test. Still, the specimen showed distinct Hackle lines and a relatively large fracture mirror, leading to the assumption that even though the crack probably started inside that void, it still generated high velocities during the breaking process. The specimen ended prematurely since the crack was not opened up by any applied tensile forces but was probably already present at the start of the loading phase, resulting in lower flexural strength value. This skewed the determinacy of the Weibull function immensely, showing how sensitive the Weibull analysis is to minute changes in specimen’s integrity. Neglecting this point results in an overall Weibull-modulus of 22.86, which is closer to the moduli of the other groups. This is also in line with the Chi-squared-test, stating no significant connection (*p* = 0.967) of fracture mode and curing time. Anyhow, these defects are as much a part of the clinical reality of RBC’s as the polymerization process and were thus not omitted from the data. As evidenced by the Chi-squaredtest, aging had a significant effect on the fracture mode. The second step of aging induced mostly thermal stress as well as water swelling; the difference in temperature led to a periodical expansion and subsequent contraction of the slabs. This change in volume might have led to a degradation of smaller insignificant surface defects into entities large enough to cause critical failure. These larger deficiencies also offered more room for water swelling to occur, since the surface directly in contact with the storage medium was increased. This is partly confirmed by the highest population of identified surface defects (corner and edge) after the second step of aging. The ISO- and 3 s-curing groups displayed consistently lower flexural strength values, manifested through their high Weibull modulus. The 10 s-group displayed a comparatively lower Weibull modulus, since the 10 s-group also had the highest variation in its identified fracture origin. Interestingly, after the last aging step, the fracture mode edge disappeared entirely while the corner reached its lowest count overall. The less-brittle fracture mode plain, however, increased strongly (see Table 3). The Weibull modulus is also representative of this trend, being the most consistent across the study. The supposed disappearance of the edge and corner defects is mostly likely because alcohol may act as a stronger plasticizing agent than water [28]. A softer surface leads to lower velocity propagating cracks, which results in less distinct fracture features (see Figure 4D) [13]. The lower values for flexural strength are indicative of this since the respective specimens took in less mechanical energy before failure. Consequently, minute surface irregularities, leading to edge and corner fractures, might not actually leave a trace of their fracture process, masking their fracture origin. Optical examination might simply not be sufficient to differentiate these plain surfaces reliably and precisely. After all, one needs to set these results into relation. In a large-scale study on the mechanical properties of nano-hybrid composites in 2012, Schmidt and Ilie reported Weibull moduli for a variety of different materials ranging from 3.6 up to 19.0 respectively [29]. The precursor study on the prototype of TePF also reported high Weibull-moduli of 14.4 as far as 20.7 [10]. The reliability of the material, used in this study, is on the higher end of that spectrum, if not the highest. This is most likely a result of the higher amount of test specimens in this study (*n* = 30 compared to *n* = 20), which reduces the uncertainty of the Weibull distribution [27].

Chemical crosslink of the resin-matrix can be measured indirectly by way of micro-indentation [30]. A distinction between two types of polymer crosslinking must be made. Chemical crosslink is generally a covalent bond between two monomers, while physical crosslink relies mostly on hydrogen bonds and van der Waal’s forces [24]. The method of crosslinking depends strongly on the monomer composition, as aromatic monomers, like Bis-GMA (Bisphenol-A-glycerolate dimethacrylate) and Bis-EMA (Bisphenol-A-ethoxylate dimethacrylate) show a lower concentration of C=C double-bonds, resulting in a lower possibility of chemical crosslink. These monomers, however, receive their strong mechanical values from a high physical crosslink. UDMA (Urethane-dimethacrylate) on the other hand has a higher concentration of C=C double-bonds, while still being able to form physical crosslink, making it a possible reaction partner for the aromatic monomers [24].

The results for the variables measured at the top surface are partly inconclusive across curing and aging conditions. Still, several slight trends could be observed.

Firstly, the mean values for all variables of each curing variation are distributed very tightly for every aging condition. This shows how effective RAFT-polymerization is at generating additional radicals through the added AFCT-agents. One can conclude that due to all curing groups being comparable for every curing mode, that RAFT-radicals partly substitute the missing photo-radicals caused by the lower amount of RE. The results of the post-hoc tests do, however, show small but significant differences due to curing. These significant differences must be interpreted with great care, since the differences in mean value are very small, which is also evidenced by the small η_p_^2^-values for the hardness values and indentation modulus. For example, the largest difference for Martens’ hardness, a great indicator of overall hardness since it incorporates both elastic and plastic deformation, was identified for the 3 s- and ISO-curing groups after aging in alcohol and amounted only to 3.8% (625.6 N/mm^2^ for the ISO-group and 601.8 N/mm^2^ for the 3 s-group). The corresponding differences for the other measured values of the aforementioned combination are equally low (ΔY = 2.5%; ΔHV = 4%; ΔCr = 3%). It is highly unlikely that these minute differences in the polymer network will have any clinical implications for the top surface.

Secondly, in contrast to the trends observed in the macro-mechanical properties, the measured variables were seemingly positively affected from step two to step three aging as evidenced by the slightly higher values, in terms of hardness and indentation modulus for all curing groups. Yet, the indentation moduli after aging step three were still lower than the initial values. Creep on the other hand remained mostly unchanged for the mentioned aging progression. As previously noted, thermocycling not only leads to an expansion and contraction of the test specimens, but also increased water sorption. The temperature-induced rise in free volume and pore size facilitates a larger amount of water entering that free volume [21], as well as formation of physical bonds, like hydrogen bonds and van der Waals forces, with the polymer chains [11,21,24], resulting in a slight increase in volume for each specimen, which potentially reduces chain-to-chain interactions and thus decreases the micro-mechanical properties [21]. The successive immersion in ethanol, as formerly stated, acts as a strong solvent of the less-densely crosslinked areas of the polymer network [11,21]. Any unreacted monomer as well as residual water may be eluted by the solvent, leading to a reduction in volume [21]. This in turn leads to an increase in density of the polymer by contracting the intermolecular spaces, caused by any of these unbound molecules, leading to relatively higher measurements of the mentioned variables. Yet, the crosslink density of the polymer-matrix did not increase, as indicated by Creep remaining almost unchanged from step two to step three aging. Furthermore, the increase in standard deviations signalizes that the possible changes in polymer structure and thus the measured mean values varied more frequently. This hints at the fact that some areas of the specimens were affected with greater severity, while others remained comparatively unharmed. Still, barring all these indications regarding the changes in polymer crosslink due to curing variation, the top surfaces showed great resistance to the successive aging steps, which is also supported by the η_P_^2^-values.

Diversely, the bottom values paint a clearer picture. The bottom surface of the ultra-fast cured specimen’s saw the sharpest decrease (or increase in terms of Creep) after the final aging condition. At the start, the differences in mean were relatively small but significant between the groups, with the curing modes forming a heterogeneous sub-set for all variables except HV. Again, these subsets have to be interpreted with care, but they did offer a hint at differences in the polymer-matrix of the different specimens. After step two aging, a trend for all sub-sets to be statistically indifferent from one another, similar to the flexural modulus and strength in the three-point bending test, was observed. The final step of aging had the greatest influence on all variables once more, as the standard deviations increased, signaling a more extreme distribution of the data most, with the most drastic rise for the 3 s-curing group. All measurements turned out to be significantly lower than their counterparts. When taking into consideration the effective differences between the curing modes, larger differences between the sub-sets were found. The percentile difference in mean for the 3 s group in comparison to the longer curing times was consistently above 10% (ISO: ΔY = 10%; ΔHM = 12%; ΔHV = 14%; ΔCr = 13%/10 s: ΔY = 13%; ΔHM = 10%; ΔHV = 10%; ΔCr = 10%). One may conclude that the 3 s-curing method led to a more heterogeneous polymer network at the bottom of the specimens in comparison to the other methods, as evidenced by the high standard deviation of that data. Not only was the polymer structure more diverse, it was also more susceptible to the effect of the ethanol solution, since, unlike the longer curing times, none of the hardness values and indentation modulus increased from step two to three aging. The suspected amount of higher chemical crosslink density at the bottom, produced by the longer curing conditions, offered a higher resistance to the final solvent. Hence, the integrity of the ISO- and 10-s-cured specimen’s surface was affected in fewer areas, evidenced by their comparatively lower standard deviations. However, the strong increase in standard deviation as well as the significantly lower values for all variables (or higher in case of Creep) led to the assumption that the ultra-fast curing mode effectively caused a substantially more heterogeneous network than its counterparts.

Microgel agglomerates are typical features of heterogeneous polymer networks and form more frequently when aromatic monomer molecules like Bis-GMA and Bis-EMA, which are part of the resin-composition of TePF, with strong hydrogen-bonds and other forms of physical crosslink are utilized [24]. The same monomers also form fewer chemical crosslinks, covalent bonds between polymer chains, resulting in a longer chain length of that polymer [24]. Pairing this with the potential increase in chain-length, caused by the decrease of RE in the ultra-fast curing modes, results in a network more prone to degradation and solvent swelling [21,24]. This was partially compensated by the LCU packing the photons more densely in a given area, as seen in the increase in absolute irradiance of the ultra-fast curing mode. Scattering of light due to differences in refractive indices is highest at the start of curing, since the translucency of a composite increases with the degree of polymerization as it transitions from a gel phase to a glassy state [31]. Additionally, a higher number of photons being emitted in a fixed timeframe also increases the chance of photon collision, further increasing light-scattering, which in turn decreases crosslink and increases the formation of microgels.

To sum up, longer curing times still lead to preferable macro- and micro-mechanical properties, through a higher level of crosslinking of the polymer-matrix as a result of more initiating photo-radicals. Yet, considering the limitations of this study, results of RAFT-modification of this bulk-fill composite are commendable. Ultra-fast curing resulted in values very close to the optimum attainable via ISO-curing. Yet, many soft factors must be considered when talking about reduced curing and in turn treatment times. Faster light-curing quickens the treatment process, leaving less room for contamination of the restoration through saliva and aerosol, when the composite is most vulnerable. Since faster light-curing is also more sensible to mishandling of the LCU, more exposure also leads to more exposure lost due to possible change of angulation [12]. This could shift the attention of the dentist more towards light-polymerization, making it a “matter of the boss” due to the importance of correct handling to guarantee adequate treatment.

After all null-hypotheses 1–3 were rejected, since aging did significantly alter the composite’s properties, optimal curing (i.e., ISO-curing) still produced superior values in comparison to the clinical (10 s-, 3 s-curing) approach, and the fracture mechanism was significantly shifted by accelerated aging. The last null-hypothesis can, however, be accepted since curing did not change the mode of fracture.

## 5. Conclusions

The variation in curing conditions for this RAFT-modified bulk-fill resin-based composite produced comparable values for both three-point bending test and depth-sensing indentation test after 24 h storage in artificial saliva and thermocycling. Differences were made clear after the storage in the ethanol solution, exposing a possible decline in polymer crosslinking caused by ultra-fast polymerization, especially at the bottom portions of the specimens, resulting in generally lower mechanical values. The reliability of the material was comparable, regardless of curing time, as evidenced by the Weibull analysis and the changes in fracture mechanism being more dominantly induced by accelerated aging. Nonetheless, shorter curing times may help to eliminate non-material-dependent factors, like contamination of the restoration and indistinct attention to the curing process, by elevating the importance of light-curing in dental practices.

## Figures and Tables

**Figure 1 materials-13-05350-f001:**
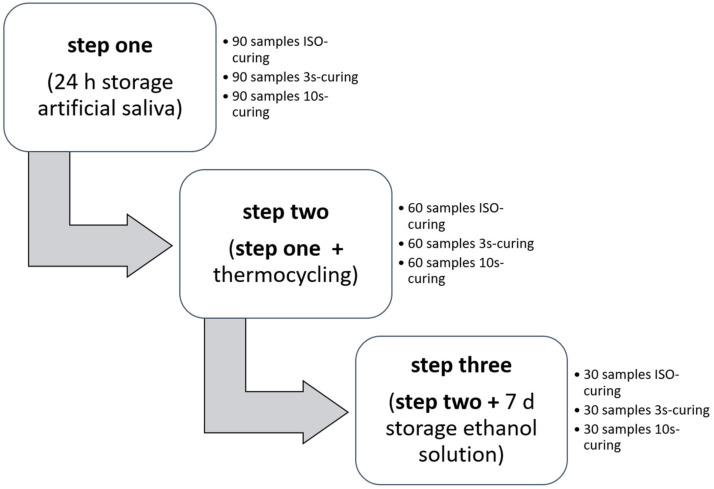
Flow chart of the in-vitro aging progression.

**Figure 2 materials-13-05350-f002:**
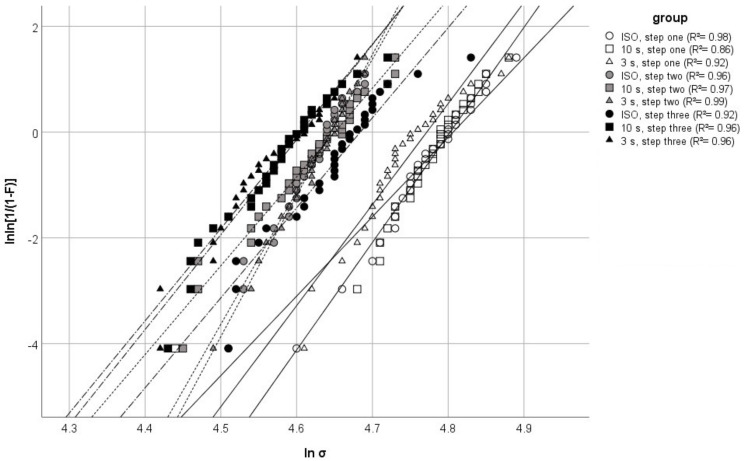
Weibull distribution as a function of curing and aging.

**Figure 3 materials-13-05350-f003:**
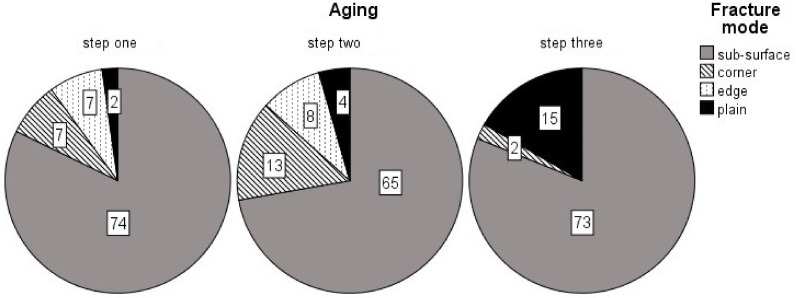
Circle chart of the observed fracture modes after each aging condition.

**Figure 4 materials-13-05350-f004:**
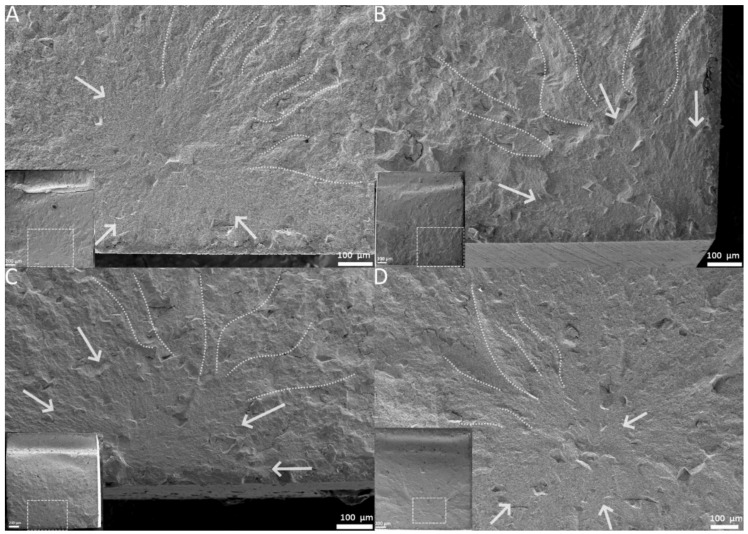
SEM images: (**A**) = sub-surface fracture origin, ISO-cure, step three aging; (**B**) = corner fracture origin, 10 s-cure, step one aging; (**C**) = edge fracture origin, 3 s-cure, step two aging; (**D**) = plain fracture without a distinct fracture origin, 10 s-cure, step three aging; lower left corner gives an overview over each specimen, dotted box shows the magnified area of the specimen; arrows identify parts of the fracture mirror, dotted lines mark Hackle lines.

**Table 1 materials-13-05350-t001:** Three-point bending test: flexural strength σ_f_ in MPa (mean and standard deviation) and Weibull modulus m with 95% confidence interval. Superscript letters indicate homogeneous groups; Capital letters indicate sub-groups related to columns; Lowercase letters indicate sub-groups related to rows; Tukey’s post-hoc test (α = 0.05).

Curing/Aging	Step One		Step Two		Step Three	
	σ_f_	m	σ_f_	m	σ_f_	m
**ISO**	118.6 ^A,a^ (7.1)	20.3 [19.3;21.4]	101.8 ^A,b^ (4.9)	25.1 [23.1;27.1]	104.9 ^A,b^ (7.4)	17.0 [15.1;18.9]
**10 s**	118.1 ^A,a^ (8.3)	15.0 [12.6;17.4]	101.4 ^A,b^ (7.3)	16.8 [15.7;17.9]	97.4 ^B,b^ (6.6)	18.0 [16.5;19.4]
**3 s**	114.9 ^A,a^ (7.2)	19.1 [16.9;21.3]	102.0 ^A,b^ (4.7)	26.7 [25.6;27.8]	97.0 ^B,c^ (6.7)	17.5 [16.1;18.8]

**Table 2 materials-13-05350-t002:** Three-point bending test: mean flexural modulus E_f_ in MPa (mean and standard deviation). Superscript letters indicate homogeneous groups; Capital letters indicate sub-groups related to columns; Lowercase letters indicate sub-groups related to rows; Tukey’s post-hoc test (α = 0.05).

Curing/Aging	Step One	Step Two	Step Three
**ISO**	7.6 ^A,a^ (0.3)	6.9 ^A,b^ (0.3)	5.9 ^A,c^ (0.3)
**10 s**	7.3 ^B,a^ (0.4)	6.9 ^A,b^ (0.3)	5.5 ^B,c^ (0.4)
**3 s**	7.1 ^C,a^ (0.2)	6.8 ^A,b^ (0.4)	5.3 ^B,c^ (0.4)

**Table 3 materials-13-05350-t003:** Depth-sensing indentation test: indentation modulus Y in GPa (mean and standard deviation). Superscript/subscript letters indicate homogeneous groups; Capital letters indicate sub-groups related to columns; Lowercase letters indicate sub-groups related to rows; Tukey’s post-hoc test (α = 0.05).

Curing/Aging	Step One		Step Two		Step Three	
	Top	Bottom	Top	Bottom	Top	Bottom
**ISO**	16.3 ^A,a^ (0.4)	16.2 _A,a_ (0.4)	14.7 ^B,c^ (0.8)	14.2 _B,b_ (0.5)	15.8 ^A,b^ (0.8)	14.7 _A,b_ (1.5)
**10 s**	15.8 ^B,a^ (0.4)	15.6 _B,a_ (0.3)	14.6 ^B,b^ (0.5)	14.6 _A,b_ (0.5)	15.6 ^A,a^ (1.0)	15.0 _A,b_ (1.2)
**3 s**	15.7 ^B,a^ (0.5)	15.3 _C,a_ (0.5)	15.2 ^A,b^ (0.3)	14.4 _AB,b_ (0.5)	15.4 ^A,ab^ (1.1)	13.3 _B,c_ (2.1)

**Table 4 materials-13-05350-t004:** Depth-sensing indentation test: Mean Martens’ hardness HM in N/mm^2^ (mean and standard deviation). Superscript/subscript letters indicate homogeneous groups; Capital letters indicate sub-groups related to columns; Lowercase letters indicate sub-groups related to rows; Tukey’s post-hoc test (α = 0.05).

Curing/Aging	Step One		Step Two		Step Three	
	Top	Bottom	Top	Bottom	Top	Bottom
**ISO**	604.2 ^A,b^ (14.2)	604.9 _A,a_ (10.2)	585.1 ^B,c^ (21.3)	566.7 _A,b_ (14.8)	625.6 ^A,a^ (32.6)	594.8 _A,a_ (56.7)
**10 s**	599.1 ^A,b^ (17.2)	595.0 _B,a_ (13.7)	580.6 ^B,c^ (17.4)	568.8 _A,b_ (19.5)	619.6 ^AB,a^ (34.7)	583.6 _A,ab_ (53.0)
**3 s**	598.4 ^A,a^ (16.7)	584.7 _C,a_ (18.7)	595.3 ^A,a^ (15.3)	569.9 _A,a_ (21.8)	601.8 ^B,a^ (51.2)	525.8 _B,b_ (83.0)

**Table 5 materials-13-05350-t005:** Depth-sensing indentation test: mean Vickers’ hardness HV in N/mm^2^ (mean and standard deviation). Superscript/subscript letters indicate homogeneous groups; Capital letters indicate sub-groups related to columns; Lowercase letters indicate sub-groups related to rows; Tukey’s post-hoc test (α = 0.05).

Curing/Aging	Step One		Step Two		Step Three	
	Top	Bottom	Top	Bottom	Top	Bottom
**ISO**	77.6 ^A,b^ (2.0)	78.1 _A,a_ (1.3)	76.3 ^AB,b^ (2.3)	74.1 _A,b_ (1.9)	81.5 ^A,a^ (4.2)	78.7 _A,a_ (7.8)
**10 s**	77.3 ^A,b^ (2.2)	77.1 _A,a_ (2.0)	75.7 ^B,b^ (2.2)	73.5 _A,b_ (2.5)	80.6 ^AB,a^ (4.4)	75.5 _A,ab_ (7.0)
**3 s**	77.2 ^A,a^ (2.4)	74.6 _B,a_ (2.5)	77.1 ^A,a^ (2.1)	74.0 _A,a_ (3.5)	78.2 ^B,a^ (6.9)	67.8 _B,b_ (11.4)

**Table 6 materials-13-05350-t006:** Depth-sensing indentation test: mean Creep Cr in % (mean and standard deviation). Superscript/subscript letters indicate homogeneous groups; Capital letters indicate sub-groups related to columns; Lowercase letters indicate sub-groups related to rows; Tukey’s post-hoc test (α = 0.05).

Curing/Aging	Step One		Step Two		Step Three	
	Top	Bottom	Top	Bottom	Top	Bottom
**ISO**	3.0 ^C,b^ (0.1)	2.9 _C,b_ (0.1)	3.4 ^A,a^ (0.2)	3.3 _B,a_ (0.1)	3.4 ^A,a^ (0.1)	3.3 _B,a_ (0.1)
**10 s**	3.1 ^B,b^ (0.2)	3.1 _B,b_ (0.1)	3.4 ^A,a^ (0.2)	3.4 _B,a_ (0.1)	3.4 ^A,a^ (0.1)	3.3 _B,a_ (0.1)
**3 s**	3.2 ^A,c^ (0.1)	3.5 _A,b_ (0.2)	3.4 ^A,a^ (0.2)	3.5 _A,b_ (0.1)	3.3 ^B,b^ (0.1)	3.8 _A,a_ (0.3)

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
