# Peer review of "Long-Term Stability of a RAFT-Modified Bulk-Fill Resin-Composite under Clinically Relevant versus ISO-Curing Conditions"

_materials, 2020, doi:10.3390/ma13235350_

Round 1

Reviewer 1 Report

Authors considered RAFT-polymerized BF-RBC, Tetric PowerFill(TePF), for mechanical properties in accordance to accelerated aging and curing methods. Experiments are thorough but there are few concerns;

  1. Introduction - Current writing outlined RAFT as new method of polymerization and TePF as RAFT applied material. Then, authors stated need of testing in accelerated aging (long term clinical environment). However, it is difficult to deduce how null hypotheses were derived from these backgrounds, especially with optimal curing (ISO method?). What is ISO method? Why authors considered ISO versus clinically relevant condition? Is authors stating that ISO is not clinically relevant? Background seems to be focused with polymerization methods, and would this mean authors have to consider RAFT based polymerized BF-RBC compare to conventional? Also, consider how null hypotheses were presented in other manuscripts.
  2. Materials and Methods
    1. Authors used aging condition of 24 hours storage in artificial saliva followed by thermocycling and then to ethanol/distilled water. Did authors immersed sample in saliva during thermocycling (ie. 4 degree celcius in salvia and 55 degree celcius in saliva etc)? 10,000 cycles shall be converted into relevant clinical days of usage. Also, how ethanol/distilled water would be a clinically relevant condition?
    2. As the comparison between ISO method and clinically relevant method would be the key to this manuscript, it would be helpful for readers for more in depth information on ISO method. In which part of ISO 4049:2019 elaborate this method (it is 2019 version by the way, and not 2018. Same for DIN version).
    3. ISO 4049:2019 would require producing of samples with 25 mm x 2 mm x 2 mm for flexural strength. Also, distance between support would be 20 mm. Would there be a reason for different dimensions? Would this influence curing method? Also, please state equations used to calculate MPa.
  3. Results: Many of the results are expressed in table which sometime is difficult to compare between groups. Perhaps expressing as bar graphs would be helpful.
  4. Discussions: Null hypotheses were not numbered in Introduction, and therefore the last sentence regarding "null-hypothese 1-3" may need revision.

Author Response

Cover Letter- Reviewer 1

All comments to the corresponding author have been addressed independently below. The author’s rebuttal is always in BLUE.

The author would firstly like to thank the reviewers’ for taking the time to read and critically appraise the manuscript and secondly to thank the reviewers for their positive constructive comments in improving the work.

Comments and Suggestions for Authors

Authors considered RAFT-polymerized BF-RBC, Tetric PowerFill(TePF), for mechanical properties in accordance to accelerated aging and curing methods. Experiments are thorough but there are few concerns;

  1. Introduction - Current writing outlined RAFT as new method of polymerization and TePF as RAFT applied material. Then, authors stated need of testing in accelerated aging (long term clinical environment). However, it is difficult to deduce how null hypotheses were derived from these backgrounds, especially with optimal curing (ISO method?). What is ISO method? Why authors considered ISO versus clinically relevant condition? Is authors stating that ISO is not clinically relevant? Background seems to be focused with polymerization methods, and would this mean authors have to consider RAFT based polymerized BF-RBC compare to conventional? Also, consider how null hypotheses were presented in other manuscripts.

Our response: Thank you for the above-mentioned observations.

As stated in lines 61 and following, both accelerated aging and variation in curing time alter a dental composite mechanical stability and morphological features. The fractographic examination of the specimen’s serves as a way of connecting the measured values and the calculated reliability with illustrative properties. The null hypotheses either evaluate the effects of aging on measured and calculated values (first null hypothesis), the effects of curing variation on measured and calculated values (second null hypothesis) or effects of aging and the variation of curing time on the morphological (illustrative) features of the specimens (third and fourth null hypothesis).

Please note that the nomenclature “ISO-method” was not mentioned in the introduction section. It is referred to as “optimal curing, as stipulated in standards”. The purpose of a standard by definition is to eliminate as many potential inaccuracies during the production and testing process as possible. As such the curing time is increased drastically, above clinical realities, to guarantee an optimal amount of cure. Additionally, the specimens are also cured directly from the bottom. A direct resin restoration, situated in a tooth, cannot be cured directly from the bottom due to anatomical limitations. A standardized curing method is thus not achievable in a clinical environment.

The introduction focuses on the ongoing changes in bulk fill resin-based composites. Conventional bulk-fill composites, as well as the predecessor of TePF, have been thoroughly investigated, as we note in lines 57-59. We also make sure to compare our results in the discussion part of the paper to studies on the the conventional bulk-fill composites in lines 336-337 and lines 394-398.

We believe that the clear-cut presentation of displaying the null hypotheses in this way helps to clarify the aim of the study, since especially fractography is not a quantitative but a qualitative approach. This also helps to show the limitations of the study.

  1. Materials and Methods
  2. Authors used aging condition of 24 hours storage in artificial saliva followed by thermocycling and then to ethanol/distilled water. Did authors immersed sample in saliva during thermocycling (ie. 4 degree celcius in salvia and 55 degree celcius in saliva etc)? 10,000 cycles shall be converted into relevant clinical days of usage. Also, how ethanol/distilled water would be a clinically relevant condition?
  3. As the comparison between ISO method and clinically relevant method would be the key to this manuscript, it would be helpful for readers for more in depth information on ISO method. In which part of ISO 4049:2019 elaborate this method (it is 2019 version by the way, and not 2018. Same for DIN version).
  4. ISO 4049:2019 would require producing of samples with 25 mm x 2 mm x 2 mm for flexural strength. Also, distance between support would be 20 mm. Would there be a reason for different dimensions? Would this influence curing method? Also, please state equations used to calculate MPa.
  5. Results: Many of the results are expressed in table which sometime is difficult to compare between groups. Perhaps expressing as bar graphs would be helpful.
  6. Discussions: Null hypotheses were not numbered in Introduction, and therefore the last sentence regarding "null-hypothese 1-3" may need revision.stronger.

Our response:

  1. Specimens were immersed in distilled water during the thermocycling process, as stated in line 100 and 101. In literature 10.000 cycles are described as approximately one year of clinical service. We chose distilled water as the storing medium since water evaporates and condensates frequently in the process. These changes would significantly impact the concentration of the artificial saliva solution if it were used as the thermocycling medium. The reproducibility of our experiments would thus not be guaranteed.

Heintze et al. showed a significant correlation between clinical index and decrease in flexural strength for the combination of thermocycling and storage in ethanol. Which had great part in conceptualizing the aging conditions. As such we made sure to reference this study (#19) in lines 295-300, to emphasize the study conceptualization.

  1. We thank you for spotting that typo in the release date of the ISO-norm. We made sure to date the ISO-norm correctly.

We made sure to accentuate the ISO-curing mode in line 83. Information about the standardized curing method is provided in lines 83 as well as in reference #14 on page 17, i.e. “irradiate that section of the specimen for the recommended exposure time”. As such, the manufacturer recommends a maximum of 20s of cure. The ISO-method serves a standard in this paper, the intent of the paper is not to discuss the applicability of the ISO-method. However, we hinted at the relevance of ISO vs clinical in lines 64-65 and lines 495-496.

We state that our three-point bending test procedure confers to NIST No. 4877, since first we made sure to adhere to the studies mentioned in the meta-analysis by Heintze et al. Secondly our study group has more than 20 years of experience combined of this way of testing and a consequent database. Also please note, that the ISO-method propagates a minimum of five testing specimens, which in our opinion would neither be representative, nor would it provide significant statistical analysis.

As stated in DIN 4049 on page 17 the reduced length of each specimen results in less overlapping irradiation sections, which does not affect the cross-sectional surface of the specimens.

Based on your recommendation, we added the equation for flexural strength and made sure to explain how the flexural modulus was calculated from this.

  1. We replaced Table 3 with a circle-chart, to visualize the results of the fractographic analysis. Still, displaying each measured variable in a bar graph, would result in at least seven bar graph charts figures, with a minimum of nine individual panels of bar charts for each combination. This would make the paper seem cluttered while also confusing the reader. Splitting these graphs for each aging or curing condition would result in artificially lengthened paper, of which about three pages would solely be figures. The less illustrative approach of a Table was considered by the authors and as such great care was taken in the indices, generated in the post-hoc test, i.e. significant differences in columns were noted in capital letters and in lowercase letters for rows. Regarding the results of the depth-sensing indentation test, values measured at the top received top-indices, while values measured at the bottom received bottom indices.
  2. Thank you for this observation, we made sure to number the null hypotheses accordingly.

Reviewer 2 Report

The manuscript entitled “Long-term stability of a RAFT-modified bulk-fill resin-composite under clinically relevant versus ISO-curing conditions” is within the scope of the Journal. The research topic is innovative, even considering the evaluation of a commercial resin composite restorative material. The search, development and evaluation of new technologies to accelerate curing time without compromising the depth of cure and the mechanical properties of resin composite is still a challenge. The hypotheses were well established, and the methods used to test them were appropriate.

Only some minor  issues were raised as follow:

- Recommendation: please state clearly the dependent and independent variables at the beginning of Materials and Methods.

- Please be consistent with keeping spaces between numbers and units throughout the manuscript.

- What was the rationale of choosing the curing protocol (curing time and curing direction)? Why did the authors not include 20 s  at the top only? Or included the curing direction in the 10 s top and 3 s top?

- How did the authors decide the specimens slabs dimensions of 16 mm x 2 mm x 2 mm?

- What were the criteria to choose the specimens for SEM fracture analysis?

- Concerning the depth-sensing indentation test, why the distribution of indentations was randomly distributed? Moreover, how the randomness was set?

- Please, describe how the sample size was calculated.

- For better comprehension and details, the statistical analysis could be described according to each dependent variable.

- In the results section, please check the bold/non-bold fonts in the tables.

- It is satisfying to see Marten’s and Vickers’ hardness been addressed.

- Lines 240-244: please, clarify the RE. Do 10.24 J correspond to the RE delivered when 3 s top and 55.29 J to the RE when ISO-curing curing protocols were employed?

- Besides being quite long the discussion, it is rich, and the authors addressed every result.

Therefore, the recommendation is ACCEPT with minor changes.

Author Response

Cover Letter- Reviewer 2

All comments to the corresponding author have been addressed independently below. The author’s rebuttal is always in BLUE.

The author would firstly like to thank the reviewers’ for taking the time to read and critically appraise the manuscript and secondly to thank the reviewers for their positive constructive comments in improving the work.

The manuscript entitled “Long-term stability of a RAFT-modified bulk-fill resin-composite under clinically relevant versus ISO-curing conditions” is within the scope of the Journal. The research topic is innovative, even considering the evaluation of a commercial resin composite restorative material. The search, development and evaluation of new technologies to accelerate curing time without compromising the depth of cure and the mechanical properties of resin composite is still a challenge. The hypotheses were well established, and the methods used to test them were appropriate.

Only some minor  issues were raised as follow:

- Recommendation: please state clearly the dependent and independent variables at the beginning of Materials and Methods.

Our response: Thank you for this observation, we made sure to state the independent and dependent variables at the beginning of section 2.6 (lines 149-151).

- Please be consistent with keeping spaces between numbers and units throughout the manuscript.

Our response: The spacing issue was fixed – please refer to the updated manuscript.

- What was the rationale of choosing the curing protocol (curing time and curing direction)? Why did the authors not include 20 s  at the top only? Or included the curing direction in the 10 s top and 3 s top?

Our response: The goal of the curing protocols was to first produce optimally cured specimens (ISO-method) in order to established a quasi gold-standard for this BF-RBC to relate the clinical protocols to. Clinicians will mostly stick to the manufacturer specified curing times, which is why we chose to omit the 20 s top only method. The curing direction of the 10 s and 3 s was not changed since in clinical reality curing can only happen from the top of the restoration not the bottom.

- How did the authors decide the specimens slabs dimensions of 16 mm x 2 mm x 2 mm?

Our response: Firstly, our study group has a combined experience of more than 20 years of producing three-point bending test specimens (as well as a consequent database) in this fashion. Secondly the ISO-method only propagates a minimum of five testing specimens, which would in our opinion would not be representative nor suited for Weibull analysis. Thirdly, we wanted to adhere the preparation protocol Heintze et al. described in their meta-analysis in 2017, which showed a significant correlation between specimens produced in this fashion and thermocycling followed by storage in ethanol. We made sure to emphasize this in lines 295-300.

- What were the criteria to choose the specimens for SEM fracture analysis?

Our response: Representative specimens for each fracture mode, i.e. they showed a distinct fracture mirror and hackle-lines etc., were chosen, based on the light-microscopic photo-documentation. The selection was then further inspected and again specimens were singled out. The final selection was then sputtered and three specimens for each fracture modes were analyzed. Of these three specimens one was then chosen for Figure 4. The goal was to provide to a “no-doubt” illustration of each fracture mode.

- Concerning the depth-sensing indentation test, why the distribution of indentations was randomly distributed? Moreover, how the randomness was set?

 Our response: We chose to select the indentation locations randomly, since choosing only visually acceptable areas for indentation would add a subjective bias to the measurements. The operator made sure to rapidly select the measurement locations, while sometimes increasing, decreasing or keeping the distance from the last location.  

- Please, describe how the sample size was calculated.

Our response: The sample sizes was not calculated, we chose n=30 as the sample size (which is way higher as requested), since we made sure to provide enough specimens for an adequate Weibull-analysis (see Ref. #27) and statistical analysis. No specimens were omitted from the statistical analysis.

- For better comprehension and details, the statistical analysis could be described according to each dependent variable.

Our response: The statistical tests applied to the measured results are standard scientific practice, we believe that further description would not serve to enhance the general discussion of the topic.

- In the results section, please check the bold/non-bold fonts in the tables.

Our response: We fixed this oversight – please refer to the updated manuscript.

- It is satisfying to see Marten’s and Vickers’ hardness been addressed.

Our response: Thank you for this positive feedback.

- Lines 240-244: please, clarify the RE. Do 10.24 J correspond to the RE delivered when 3 s top and 55.29 J to the RE when ISO-curing curing protocols were employed?

Our response: Based on your observation we made sure to specify the irradiation surface for the curing protocols in lines 254-257.

- Besides being quite long the discussion, it is rich, and the authors addressed every result.

Our response: Thank you for this observation.

Reviewer 3 Report

Long-term stability of a RAFT-modified bulk-fill resin composite under clinically relevant versus ISO-curing conditions

MANUSCRIPT NUMBER: materials-993089

The aim of the present investigation was to evaluate to investigate the stability of a RAFT-modified bulk-fill resin composite in a in vitro study model.

GENERAL COMMENTS

The study is original and interesting due to the modern orientation of dental materials. The investigation methodologies are appropriate for the paper topic. The discussion section is well written according to the recent bibliography. According to the following comments, the present paper is suggested for publication after minor revision

Abstract

The aim of the study is missed in the section.

Methods

What was the different light curing mode selected? (ex. continuous curing, pulse delay, soft-start or intermittent light?) More images of the in vitro simulation should be added to the main text if available.

Discussion

The limitation of the study be added to the present section.

Author Response

Cover Letter- Reviewer 3

All comments to the corresponding author have been addressed independently below. The author’s rebuttal is always in BLUE.

The author would firstly like to thank the reviewers’ for taking the time to read and critically appraise the manuscript and secondly to thank the reviewers for their positive constructive comments in improving the work.

Comments and Suggestions for Authors

 GENERAL COMMENTS

The study is original and interesting due to the modern orientation of dental materials. The investigation methodologies are appropriate for the paper topic. The discussion section is well written according to the recent bibliography. According to the following comments, the present paper is suggested for publication after minor revision

Abstract

The aim of the study is missed in the section.

Our response: Thank you for this observation - The aim of the study was more clearly formulated in the abstract – please consider changes in the manuscript text.

Methods

What was the different light curing mode selected? (ex. continuous curing, pulse delay, soft-start or intermittent light?) More images of the in vitro simulation should be added to the main text if available.

Our response: We made sure to signal that the LCU-used only employs continuous curing in line 83. The in-vitro simulation involved storage in different solutions and the use of a common device for thermocycling. We believe that images of the in-vitro simulation (i.e. an image of the device) would not add information but clutter this clear-cut section of the paper. Additionally, they would provide no additional benefit for the actual discussion of the paper. Still, we added a flow-chart illustration of the successive aging process in line 90, to further clarify the protocol.

Discussion

The limitation of the study be added to the present section.

Our response: We made sure to remind the reader of the limitations of the study in the final section of the discussion in line 487

Round 2

Reviewer 1 Report

All of the comments are well addressed.